# Valorization of Brewer’s Spent Grain Liquid Fraction for the Development of a Pasteurized Strawberry-Based Blend Juice

**DOI:** 10.3390/foods14234053

**Published:** 2025-11-26

**Authors:** Valentina Ariela Neira Monsalve, Bastián Benjamín Devia Valenzuela, Alonso Ignacio Peña Arias, Heidi Laura Schalchli Sáez, Priscilla Siqueira Melo, Carolina de Souza Moreira, Alan Giovanini de Oliveira Sartori, Christian Arnoldo Vergara Ojeda, Severino Matias de Alencar, Erick Sigisfredo Scheuermann Salinas

**Affiliations:** 1Graduate Program Master in Engineering Science Mention Biotechnology, Faculty of Engineering and Sciences, Universidad de La Frontera, Temuco CP 4780000, Chile; v.neira02@ufromail.cl; 2Undergraduate Program Biotechnology Civil Engineering, Faculty of Engineering and Sciences, Universidad de La Frontera, Temuco CP 4780000, Chile; b.devia01@ufromail.cl; 3Undergraduate Program Chemical Civil Engineering, Faculty of Engineering and Sciences, Universidad de La Frontera, Temuco CP 4780000, Chile; a.pena11@ufromail.cl; 4Biotechnological Research Center Applied to the Environment (CIBAMA-BIOREN), Universidad de La Frontera, Temuco CP 4780000, Chile; heidi.schalchli@ufrontera.cl; 5Chemical Science and Natural Resources Department, Universidad de La Frontera, Temuco CP 4780000, Chile; 6Department of Food Science and Technology, Escola Superior de Agricultura Luiz Queiroz (ESALQ), Universidade de São Paulo, Piracicaba CEP 13418-900, Brazil; priscilla.esalq@gmail.com (P.S.M.); moreirasc1@usp.br (C.d.S.M.); alangosartori@usp.br (A.G.d.O.S.); 7Chemical Engineering Department, Universidad de La Frontera, Temuco CP 4780000, Chile; christian.vergara@ufrontera.cl

**Keywords:** Brewers’ spent grain, dewatering, liquor, thermal treatment, sensory juice

## Abstract

Brewer’s spent grain (BSG) is a high-moisture (70–80%) by-product of the brewing industry. During dewatering, it yields a liquid fraction that can be combined with strawberry pulp (SP) to create a new food product. This study evaluated the use of BSG liquid fraction as a dilution medium for SP to develop a novel blend juice (BJ) and assess its shelf-life after pasteurization. The optimal formulation consisted of 70% BSG liquid fraction and 30% SP, pasteurized at 95 °C for 5 min (BJ5) and 10 min (BJ10), and stored for 8 weeks at 25 °C. Compared to BJ, pasteurization (BJ5 and BJ10) caused significant (*p* < 0.05) changes in moisture, color (Δ*E*), soluble solids, and aerobic mesophilic bacteria. Both BJ5 and BJ10 showed sensory attributes, overall acceptance, and purchase intent comparable to two commercial Chilean blend juices. During the 8-week storage, both pasteurization conditions ensured microbiological stability, while color and other chemical parameters remained stable. Nonetheless, moisture, soluble solids, pH, and total polyphenol content were significantly affected (*p* < 0.05). Overall, the BSG liquid fraction can be successfully repurposed to create a microbiologically stable, sensorially appealing strawberry blend juice with strong potential for the food and gastronomy industries.

## 1. Introduction

Brewer’s spent grain (BSG) is a by-product of the brewery industry, composed of malt grain husks. Its annual worldwide production is around 37 million tons, which is usually discarded or used for animal feed [1,2,3].

BSG is rich in dietary fiber and protein. It also contains other important nutrients and bioactive compounds, such as lipids, starches, sugars, minerals, vitamins, and polyphenolic compounds [1,2,4]. With a moisture content of 70–80%, BSG often requires dewatering to stabilize the solid fraction [5,6,7,8]. The resulting liquid fraction has shown potential for various applications, including developing plant-based yogurt alternatives [9] and serving as a fermentation medium [10,11].

Using a Type 3A Davenport press to dewater wet BSG (from malt and corn), Finley et al. [5] reported a press water yield of 42.1%, which contained 3% solids. El-Shafey et al. [6] reported that the moisture of BSG cakes was reduced from 80 to 51% through a filtration stage involving hot-water squeezing, and finally to 20–30% by applying vacuum to the hot-squeezed cakes. Using a similar method, Machado et al. [7] removed up to 95% of the water from the BSG, thereby decreasing its moisture content from 75 to 15%. Bjerregaard et al. [12] used a novel continuous rotary drum press, yielding a liquid filtrate equivalent to 50% of the hot BSG mass. Treating BSG with and without ultrasound, followed by pressing in a 2 L stainless steel manual press, yielded 53.7 and 47.2 mL of liquid fraction per 100 g of BSG, respectively. This process reduced the BSG moisture from 71.2–72.4 to 59.1–59.8% [8]. Akermann et al. [10] reported a yield of ∼17.6 kg of liquid fraction (liquor) per ∼34 kg of wet BSG, reducing its moisture content from ∼75 to ∼65 wt%. While these studies focus on dewatering the solid BSG, the resulting liquid fraction is often overlooked. The disposal or use of the BSG liquid fraction should be studied to mitigate adverse environmental effects [3,8].

The composition of the BSG liquor obtained from *Wheat bock*, *Wheat*, and *Helles* brewing recipes by using a friction press (ENOL OP 20, Wein GmbH) includes sugars, protein, amino acids, and minerals [10]. According to Shetty et al. [11], the BSG liquid produced using a rotary drum press equipped with a 300 μm filter contained ca 12 g L^−1^ of fermentable (maltose and glucose) and unfermentable (raffinose and maltodextrin) sugars. In addition, small amounts of lactic acid (1.2 g L^−1^), acetic acid, and citric acid were also observed. Madsen et al. [9] filtered a 100 µm liquid fraction from BSG, which was blended with a commercial unsweetened soy drink in a ratio of 20:80, and later used to develop a product similar to yogurt. The former was chemically characterized by the presence of sugars, mannitol, pectin, acids, and ethanol. D-erythrose, fructose, D-tagatofuranose, xylose, glucopyranose, maltose, D-turanose, and cellobiose [13] were the main sugars found in the BSG water extract.

Strawberry pulp is a popular ingredient used in juices, yogurts, jams, jellies, and bakery products due to its desirable sensory characteristics, such as its vibrant color and distinct aroma, as well as its rich profile of bioactive compounds [14,15,16]. Blends of different juices, pulps, purees, or extracts, including those obtained from strawberry, have been studied to evaluate changes in their physical, chemical, and sensory characteristics and to become more attractive to consumers who seek bioactive compounds [17,18,19,20,21].

Recently, two processes have been proposed for transforming brewer’s spent grain into powders using pressing to separate the liquid and solid fractions, both with potential uses in food applications [8]. Building on this concept, we considered that the liquid fraction obtained from BSG dewatering, combined with strawberry pulp, could contribute to its physical and chemical properties to create a novel blend juice that can be effectively preserved through pasteurization. Therefore, this study aimed to evaluate the BSG liquid fraction as a dilution medium for strawberry pulp to produce a new blend juice and to assess its shelf-life after pasteurization.

## 2. Materials and Methods

### 2.1. BSG Liquid Fraction, Strawberry Pulp, and Blend Juice

The wet BSG, with a moisture content of 72.34% in weight base, fat 1.65%, protein 5.13%, crude fiber 3.19%, ash 0.68%, nitrogen-free extract 17.01%, and sodium 2.3 mg/100 g, was obtained from Birrell Ltda. (Villarrica, Chile), a craft brewery located in Villarrica, Chile, in the Araucanía Region. The BSG was collected by the research team directly from the craft brewery immediately after mashing, with the process temperature reaching approximately 70 °C. The only batch used throughout the study was stored in washed and sanitized plastic crates, which were only opened when filling them with BSG. Transport to the laboratory took approximately one hour. In the laboratory, the BSG was immediately portioned into 1010 g samples, packaged in Ziploc^®^ bags (26.8 × 27.3 cm, Racine, WI, USA), and immediately frozen at −18 °C. An additional 10 g of each sample was reserved for moisture analysis in every experimental replicate after thawing the BSG. This step was carried out to ensure the accuracy of the measurements required for the study. Each bag was thawed in a refrigerator for 24 h at 4–5 °C. After thawing, the BSG was pressed at room temperature (23–25 °C) for 5 min using a manual press equipped with a 2 L stainless steel container and a mechanical screw. In each repetition, 1000 g of wet original BSG was pressed to produce two fractions: liquid and solid. The BSG liquid fraction (BSG-LF) was stored in Schott glass bottles and frozen at −18 °C [8].

The strawberry pulp (SP) was obtained as a frozen product from Varfel (https://www.varfel.cl/, accessed on 14 October 2025), located in Temuco, Chile, which is a producer of natural and commercial frozen pulp from various fruits and vegetables.

Before preparing the blend juice, both frozen components were thawed in a refrigerator for 24 h at 4–5 °C.

The optimal proportion of the blend juice (BJ) was determined during preliminary experiments as 70% BSG liquid fraction and 30% strawberry pulp.

### 2.2. Blend Juice Pasteurization

For pasteurization, 10 mL of BJ was placed into 20 mL screw-cap test tubes (N° 9825, Pyrex^®^, Ciudad de México, México). The tubes were heated in a thermoregulated water bath (Memmert WNB 14, Schwabach, Germany), fixed at 95 °C. Temperatures were monitored with thermocouples (Lutron TM-906A, Taipei, Taiwan). One thermocouple was placed in the water bath, while a second was inserted into the geometric center of the BJ volume within a control tube. Pasteurization times were 5 min (BJ5) and 10 min (BJ10). Immediately after heating, the test tubes were cooled in an ice-water bath (~2 °C). The temperature was also monitored. For each treatment time (5 and 10 min), twenty-five tubes with blend juice were processed. The pasteurization parameters were selected based on the study by Xue et al. [22].

### 2.3. Shelf-Life of Pasteurized Blend Juice

After pasteurization, twenty tubes from each thermal treatment (BJ5 and BJ10) were stored using a temperature-controlled chamber (Archiclima, Temuco, Chile) at 25 °C for eight weeks to evaluate their shelf-life. Each tube contained 10 mL of the 70% BSG-LF—30% SP blend juice. Microbiological characteristics, as well as moisture, color, and other chemical features, were evaluated at weeks 1, 2, 4, and 8. The experiment was carried out in triplicate.

### 2.4. Analysis

#### 2.4.1. Moisture Content

Samples weighing between 4 and 5 g were used to determine the moisture content in an oven at 105 °C for 2 h, and then weighed. They were kept at 105 °C until a constant weight was achieved [8,23]. The results are reported on a wet basis (w.b.).

#### 2.4.2. Instrumental Color

The instrumental color was measured using a Minolta Chromameter CR-200b colorimeter (Osaka, Japan) with the CIE *L*a*b** system, as described by Ruíz et al. [8] and Ihl et al. [24]. The colorimeter was first calibrated using a white standard tile (Y = 93.1, x = 0.3140, y = 0.3212) under illuminant C conditions (6774 K). Color parameters were expressed using the CIE Lab system, where *L** represents lightness (0 = black, 100 = white), *a** indicates red (+*a**) to green (−*a**), and *b** denotes yellow (+*b**) to blue (−*b**).

For color measurement, each sample (20 mL) was homogeneously distributed in a 7 cm diameter Petri dish. Readings were taken by placing the instrument on the surface at five different points on each sample.

The Δ*E*, which quantifies total color differences [25], was determined by Equation (1), following the analytical classification of Adekunte et al. [26], where values indicate very different colors (Δ*E* > 3), medium differences in colors (1.5 < Δ*E* < 3) and small differences in colors (Δ*E* < 1.5).(1)$$\Delta E=\sqrt{{\Delta a*}^{2}+\Delta {b*}^{2}+\Delta {L*}^{2}}$$

#### 2.4.3. Soluble Solids

Samples were homogenized and measured with a refractometer (model 10430, Reichert-Jung, Depew, NY, USA) as °Brix [27].

#### 2.4.4. Reducing Sugar Content

The reducing sugar content was determined by the Miller method using 3,5-dinitrosalicylic acid (DNS) [13,28]. Initially, all samples were centrifuged at 10,000 rpm (10 min) using a centrifuge (Beckman GS-15, Brea, CA, USA). The resulting supernatant sample was diluted to 1/150 with distilled water. For the assay, 0.5 mL of the diluted sample was mixed with 1.5 mL of DNS reagent in a test tube. This mixture was then heated to 100 °C in a water bath for 5 min, followed by immediate cooling in an ice bath for 3 min. After cooling, 7.5 mL of distilled water was added, and the absorbance was measured at 540 nm. The concentration of reducing sugar was estimated from the standard glucose curve, and the results were expressed as g per 100 g of dry matter (d.m.).

#### 2.4.5. pH

The pH of samples was measured directly in the homogeneous mass using a pH meter. The pH meter was calibrated using buffered standards at pH levels of 4.01 and 7.00 [27].

#### 2.4.6. Titratable Acidity (TA)

Titratable acidity (TA) was determined by potentiometric titration as described by Xue et al. [22]. Briefly, a 5 mL sample was titrated with 1 mol L^−1^ sodium hydroxide solution to a final pH of 8.2 (±0.1). The *TA* content was then calculated using Equation (2) and expressed in terms of citric acid equivalent per liter.(2)$$TA\;\left(\%\right)=\frac{C_{NaOH}\cdot V_{NaOH}\cdot K}{W}\cdot 100\%$$
where *C_NaOH_* is the concentration of sodium hydroxide (1 mol L^−1^), *V_NaOH_* is the volume of sodium hydroxide (mL), *K* is the conversion factor for citric acid (0.064), and *W* is the mass of the measured sample (g).

#### 2.4.7. Total Polyphenol Content (TPC)

The total polyphenol content (TCP) was determined using the Folin–Ciocalteau method. As a preliminary step, samples were centrifuged (10,864× *g*, 10 min; Beckman GS-15, Brea, CA, USA with an F0685 fixed-angle rotor). A 40 µL aliquot of each supernatant sample was mixed with distilled water (3.16 mL) and 200 µL of Folin–Ciocalteau reagent. After 5 min, 600 µL of a 20% Na_2_CO_3_ solution was added to the mixture. Samples were kept in the dark at 20 °C for 120 min. Finally, the absorbance was measured at 765 nm using a spectrophotometer (Thermo Fisher Scientific Genesys 10, Waltham, MA, USA), and the results were expressed as mg of gallic acid equivalents (GAE) per 100 g dry matter (d.m.) [29].

#### 2.4.8. Ascorbic Acid Content

Samples were initially centrifuged (10,864× *g*, 10 min; Beckman GS-15 with an F0685 fixed-angle rotor). Subsequently, an ascorbic acid reflectometric test strip (Merck, Darmstadt, Germany) was immersed in the supernatant for 20 s. After immersion, excess liquid was removed from the strip, and the ascorbic acid concentration was measured by inserting the strip for 10 s into the Merck (Darmstadt, Germany) RQFlex reflectometer at 570/657 ± 10 nm [24]. The results were expressed as mg ascorbic acid per 100 g dry matter (d.m.).

#### 2.4.9. Microbiology Determination

The aerobic mesophilic bacteria (AMB) and the yeast and mold (YM) populations were assessed according to the Public Health Institute of Chile (Instituto de Salud Pública de Chile) [30] and Choo et al. [31]. AMB were enumerated using plate count agar (PCA), while YM counts were analyzed using potato dextrose agar (PDA) with the addition of 10% tartaric acid. The PCA plate was incubated at 37 °C for 1–2 days, whereas the PDA plate was incubated at 25 °C for 3–5 days. Results were expressed as log colony-forming units (CFU) per mL of sample.

### 2.5. Sugar Characterization by HPLC

High-performance liquid chromatography (HPLC) analysis of sugars was conducted using a Shimadzu LC-20AT system (Kyoto, Kyoto, Japan) equipped with a refractive index detector (RID) and a Shim-pack GIST NH2 column (4.6 mm × 250 mm, 5 μm). Chromatographic separation was performed under isocratic conditions with a mobile phase of acetonitrile and water (75:25, *v*/*v*) at a flow rate of 1.0 mL/min. Prior to injection, all five samples were centrifuged, and the resulting supernatants were diluted 1:1 with the mobile phase. The samples were subsequently filtered through 0.22 µm membrane filters, and 20 µL aliquots were injected into the HPLC system in triplicate. The authentic standards of sugars used in the analysis were cellobiose, xylose, maltose, glucose, fructose, sucrose, raffinose, nystose, kestose, and turanose. Sugar identification was achieved by comparing sample retention times with those of authentic standards. Moreover, quantification was performed using calibration curves prepared from standard solutions.

### 2.6. Sensory Evaluation

A sensory analysis was performed according to Lepaus et al. [32] and Cendrowski et al. [33]. The study compared three sample groups: (i) the unpasteurized 70% BSG-LF and 30% SP blend, (ii) the pasteurized blend, and (iii) two leading commercial blend juices, purchased locally in Temuco, Chile.

The evaluation was performed by 26 untrained panelists, comprising staff and students from the Department of Chemical Engineering at Universidad de La Frontera, Temuco, Chile. The panel was gender-balanced (13 females, 13 males) with ages ranging from 20 to 62 years (mean age 37.9 ± 13.4 years).

The panelists rated seven attributes (color, appearance, consistency, aroma, taste, overall acceptance, and purchase intent) using a nine-point Hedonic scale (1 = ‘dislike extremely’; 9 ‘like extremely’). Each panelist received 30 mL of each sample, served in transparent plastic cups coded with three-digit random numbers. The presentation order was randomized, and water was provided for palate cleansing between samples.

The selection of two commercial juices was based on the four following criteria: (i) national market presence, (ii) natural-like quality, (iii) high retail price, and (iv) being a fruit blend. The chosen products were a strawberry–plum berry blend juice (Guallarauco, https://www.guallarauco.cl/, accessed on 14 October 2025), listed as containing concentrated grape juice, strawberry, and plum pulp; and an apple–cherry blend juice (AFE, https://www.jugoafe.cl, accessed on 14 October 2025) listed as exclusively made from green apples and selected fresh cherries, without any additional additives.

### 2.7. Statistical Analysis

The data were subjected to analysis of variance for experiments described in Section 2.1, Section 2.2 and Section 2.3, which were carried out in triplicate. For significant differences, Tukey’s Honest Significant Difference (HSD) and Student’s *t*-test as post hoc analyses were applied. Mean values were considered significantly different at *p* < 0.05. Minitab^®^ Statistical Software 21.0.3.1.0 (Chicago, IL, USA) was used for data analysis.

## 3. Results and Discussion

### 3.1. Characterization of BSG Liquid Fraction, Strawberry Pulp, and Blend Juice

Table 1 presents the characterization of the BSG liquid fraction (BSG-LF), strawberry pulp (SP), and blend juice (BJ) in terms of moisture, color, chemical parameters, and microbiological counts (aerobic mesophilic bacteria, yeast, and molds). As expected, the characteristics of BJ were consistent with its composition, which was a mixture of 70% and 30% of BSG-LF and SP. The proximate composition of BSG-LF used in this study was as follows: fat (<0.15%), protein (0.4%), ash (<0.043%), and nitrogen-free extract (12.5%). Additionally, its sodium content was below 4 mg/100 g.

The moisture content (Table 1) of the BSG-LF obtained by pressing was 87.0% (w.b.). Although no previous reports were found on the moisture content of liquor or press water, Finley et al. [5] reported a ratio of 10,008 lb of water to 690 lb of solids after centrifugation of 10,698 lb of BSG press liquid, which corresponds to approximately 93.6% moisture. Bjerregaard et al. [12] reported solid contents of 9.45% and 10.3% in a liquid filtrate obtained from pressed hot BSG using 100 and 300 μm pore size filters on a continuous rotary drum, respectively.

The color values for *L** (43.7), *a** (0.2), and *b** (12.9) of BSG-LF are similar to or lower than those found by Ruíz et al. [8] for wet and pressed solid BSG, which were *L** 44.1 and 45.0, *a** 5.3 and 4.9, and *b** 22.5 and 22.6, respectively. According to Adekunte et al. [26] classification, which considered Δ*E* > 3 as very different colors, the Δ*E* (13.2) of BSG-LF, determined using the BJ color parameters as a reference, indicates a strong change in color after blending BSG-LF with SP, as expected.

According to some studies [10,11,13], the liquid obtained from BSG for different methods included several sugars in its composition, which are represented in levels of soluble solids (13.5 °Brix) and reducing sugars (65.6 g 100 g^−1^ d.m.) determined for BSG-LF (Table 1 and Table 2).

The pH (6.26) of BSG-LF is slightly higher than the values (5.13–5.98) reported by Akermann et al. [10] for BSG liquor obtained from pressed BSG, namely *Wheat bock*, *Wheat*, and *Helles* (Barley). The low titratable acidity (%) is in accordance with the small amount of lactic acid (1.2 g L^−1^), acetic acid, and citric acid reported by Shetty et al. [11].

The TPC in the BSG-LF (169.4 mg GAE 100 g^−1^ d.m.) shows that polyphenols present in wet BSG [1,2,4] are transferred to the liquid fraction by the pressing operation according to Ruíz et al. [8], who reported a TPC of 304–305 mg GAE 100 g^−1^ d.m. for BSG powder. Previous studies have reported a wide range of TPC values for BSG. For example, Carciochi et al. [34] found values ranging from 159 to 357 mg GAE 100 g^−1^ BSG d.m. In other studies, Meneses et al. [35] and Bonifácio-Lopes et al. [36] determined higher TPC values of 713 and 1300 mg GAE 100 g^−1^ BSG d.m., respectively. However, the TPC values of BSG liquid or liquor have not been reported before. The *Hibiscus sabdariffa* extract, used to develop blends with different fruit juices, showed a phenolic content of 1496 mg GAE 100 g^−1^ [37], which is 8.8 times higher than the TPC of BSG-LF. Defective coffee green beans and coffee silver skins as by-products, treated with 5 mL of 95% (*v*/*v*) methanol, showed a TPC of 2722 and 536 mg GAE 100 g^−1^ d.m., respectively [38]. Although polyphenols are present in BSG-LF, their total content is generally lower than in other matrices.

The analytical method used to determine ascorbic acid has a quantification range of 25–450 mg L^−1^; however, the ascorbic acid concentration in the BSG-LF was below 25 mg L^−1^, which is equivalent to a concentration of less than 16.7 mg 100 g^−1^ d.m.

Aerobic mesophilic bacteria count in the BSG-LF was 4.8 log CFU mL^−1^. This value falls within the 2.58 to 6.12 log CFU g^−1^ range reported by Robertson et al. [39] for BSG immediately after lautering, a range those authors considered as microbiologically stable and acceptable for food use. It is therefore likely that the pressing process, described in Section 2.1, could be an operation that contributes to transferring part of the AMB load from the original wet BSG obtained from the craft brewery to the BSG-LF, as noted by Ruíz et al. [8].

As shown in Table 1, strawberry pulp (SP) had a significantly lower moisture content than the BSG-LF (*p* < 0.05). Therefore, the BSG-LF allows the pulp to be diluted for consumption as juice. Strawberry juice shows a moisture of 91% according to Basiony et al. [40].

The colors of concentrated and dried strawberry pulps have been reported with values of *L** of 26.9, *a** of 19.7, and *b** of 7.3 [14], and *L** of 22–38, *a** of 19–22, and *b** of 2–10 [16], respectively, which are similar to the color parameters determined in this work for SP. The Δ*E* (8.4) of SP, determined using the blend juice color parameters as a reference, evidences a strong change in color [26] after blending SP with BSG-LF.

Additionally, the pH of our strawberry pulp (SP) aligns with the range of 3.3 to 3.7 reported by El Moutaouakil et al. [41] for several pasteurized and unpasteurized strawberry pulps. In terms of microbiological quality, our results are consistent with those of El Moutaouakil et al., who reported that after pasteurization (85 °C/7.5 min), the finished products do not contain microbiological agents such as total aerobic mesophilic flora, yeasts, molds, and enterobacteria. However, the soluble solid content (29.1 °Brix) determined for the SP, which is a concentrated product, is 2.7 to 4.1 times higher than the values (7–11 °Brix) reported by El Moutaouakil et al.

Typically, strawberry products such as fruit, pulp, and juice are known to be rich sources of several acids, polyphenols, and ascorbic acid [15]. As expected, the concentrations of these compounds are significantly higher in the SP than in the BSG-LF.

The final blend juice (BJ), formulated with 70% of BSG-LF and 30% of SP, had a moisture content of 82.4% (Table 1). This value is slightly lower than the 85.9–88.2% moisture range reported by Atef et al. [19] for several proportion blends of pumpkin juice with orange, carrot, and lemon juices, with individual moisture values of 91.4, 84.7, 89.0, and 89.8%, respectively. The moisture content of our blend agrees with the proportion volume used in its formulation, and it is significantly different from the values of BSG-LF and SP.

The soluble solids content of our blend juice (17.2 °Brix) is higher than the 7-to-13.5 range reported for several pumpkin juice blends [19]; however, their pH values (5.03–5.71) are higher than those of the BJ (pH 4.28). Feng et al. [21] report values of total soluble solid contents of 7.8 °Brix, pH 3.38, total phenols of 877.69 mg GAE L^−1^, ascorbic acid content of 2.78 mg 100 mL^−1^, total aerobic bacteria of 4.19 log CFU mL^−1^, and YM of 4.21 log CFU mL^−1^ for blended strawberry–apple–lemon juice.

In general, for most parameters in Table 1, there were significant differences (*p* < 0.05) among BSG-LF, SP, and BJ. Although strawberry pulp is the smallest proportion in the blend juice formulation, the final BJ did not differ significantly (*p* > 0.05) for the parameters *L**, *b**, and pH, which would allow the BJ to be recognized as a strawberry-derived product.

### 3.2. Effect of Pasteurization on Blend Juice

The temperature profiles for the 5- and 10- minute pasteurization treatments are presented in Figure 1 and Figure 2, including their subsequent cooling phases. The measurements were recorded at the geometric center of the sample contained in a 20 mL screw-cap test tube, as well as in the surrounding thermoregulated water bath.

As mentioned in the methodology, the water temperature of the thermoregulated bath equipment was fixed at 95 °C. When the rack with 20 tubes was introduced to the bath water, the blend juice’s temperature rapidly increased, reaching a stable plateau after 180 s at a temperature below 95 °C for both pasteurization treatments (Figure 1 and Figure 2). Xue et al. [22] reported that thermal pasteurization of 95 °C for 2 min completely inactivated the total plate count, coliforms, yeast and mold in bayberry juice. Pasteurization treatment in Figure 1 shows that the temperature stabilization was 2 min (180 to 300 s), at a range of 90–95 °C in the blend juice. In Figure 2, the period of temperature stabilization was 7 min (180 to 600 s) for the same range of 90–95 °C. These time–temperature combinations are critical for ensuring microbial inactivation. Radziejewska-Kubzdela [42] evaluated the effect of ultrasonic, thermal, and enzymatic treatment on yield and content of bioactive compounds in strawberry mash using a batch heated up to a temperature of 80 °C for 15 min and then held for 5 min at this temperature. The mash heat was equivalent to other treatments for preservation of ascorbic acid, phenolic compounds, and antioxidant activity, which in the last case was significantly higher than samples of untreated mash. This supports the potential of our shorter, higher-temperature pasteurization to maintain the nutritional quality of the blend while ensuring its safety.

Table 2 presents the sugar content determined by HPLC before and after pasteurization of the blend juice obtained from a mixture of 70% BSG-LF and 30% SP.

The sugars identified in the BSG-LF in this study have also been reported by others (Table 2). Glucose, maltose, and raffinose were detected in the BSG liquid fraction in the study by Madsen et al. [9], while maltose and glucose were identified in the liquid fraction of BSG by Shetty et al. [11].

The addition of strawberry pulp was responsible for increasing the level of reducing sugar content in the BJ. Since the BSG-LF did not contain fructose and exhibited lower glucose levels, its combination with strawberry pulp enhanced the concentrations of these monosaccharides, which may improve the palatability and energy value of the beverage. On the other hand, the BSG-LF showed higher levels of the disaccharide maltose, composed of two glucose units, and the trisaccharide raffinose, composed of galactose, fructose, and glucose. Complex carbohydrates may contribute to the prebiotic potential of the beverage, as they are less readily digested and can promote beneficial gut microbiota activity [43].

Initially, when performing a mass balance for the 70% BSG-LF and 30% SP proportion, the sugar concentrations were consistent for all those listed in Table 2. In the BJ blend, the fructose originates solely from the strawberry pulp, which explains why it decreases to approximately one-third of the initial value, considering the mass ratio. Similar decreases were observed for glucose, maltose, and raffinose. However, the concentrations of these sugars remained nearly constant after pasteurization for 5 min (BJ5) and 10 min (BJ10) when compared to the unpasteurized BJ.

The measured concentrations of fructose and glucose in the BJ were slightly, but not significantly, higher than those predicted by mass balance calculations, considering the 30% proportion of SP in the mixture. This may be due to more effective extraction of sugars from the fruit when mixed with BSG-LF. The same reasoning applies to maltose, which is lower in BJ compared to the pure BSG-LF, consistent with the 70% dilution factor. Additionally, when mixed with the pulp (SP), changes in pH and acidity (Table 1) may possibly promote a partial degradation of maltose in glucose.

Table 3 plots the effects of pasteurization at 95 °C for 5 and 10 min on blend juice (70% BSG-LF and 30% SP) in terms of its moisture, color, chemical parameters, aerobic mesophilic bacteria count, and yeast and mold count. No relevant changes were observed in most parameters, and the AMB load was eliminated by both heat treatments (Figure 1 and Figure 2).

Moisture content (Table 3) of the blend juice decreased significantly (*p* < 0.05) after pasteurization (5 and 10 min). However, the level is still very high (81.9 and 81.7%, w.b.), which, together with the nutrient composition in blend juice, creates an optimal environment for the growth of spoilage microorganisms [22].

Δ*E* values (1.7 and 1.2), determined by comparing blend juices before and after pasteurization, showed medium and small differences in color, according to Adekunte et al. [26].

The soluble solid content increased significantly (*p* < 0.05) after both pasteurization treatments. This increase may be likely due to the release of sugars and organic acids from the portion of pulp strawberry by breakdown of enzymes and temperature [31].

A significant decrease in pH (*p* < 0.05) was observed only in the BJ subjected to the 10 min pasteurization compared to the value before submitting the blend juice to this thermal treatment. This acidification could be attributed to the heat-induced release of organic acids from the strawberry pulp. Choo et al. [31] and Xue et al. [22] did not observe significant changes in the quality of noni juice (90 °C for 1 min) and bayberry juice (95 °C for 2 min) after thermal pasteurization. This behavior aligns with the pasteurization of blended juice at 95 °C for 5 min, whose pH values did not differ significantly (*p* > 0.05) from those in its fresh condition.

Similar to BJ, Choo et al. [31] did not detect yeast and mold in fresh and pasteurized (90 °C for 1 min) noni juice.

### 3.3. Sensory Evaluation of Blend Juices

Table 4 shows the sensory evaluation of BJ and two recognized Chilean commercial blend juices, before and after 5 and 10 min pasteurization at 95 °C, including acceptance and purchase intent.

Figure 3 presents the blend juice (BJ), BJ pasteurized at 95 °C for 5 min (BJ5) and 10 min (BJ10), and commercial blend juices FS and AFE, according to Section 2.6.

The sensory panel, composed of 26 untrained but balanced-gender panelists (mean age 37.9 ±13.4 years), detected significant differences (*p* < 0.05) only in the color and taste attributes among all evaluated samples (Table 3). However, pasteurization (BJ5 and BJ10) had no significant negative impact (*p* > 0.05) on any sensory attributes evaluated, overall acceptance, and purchase intent when comparing the treated juices to BJ. Our results show that the BJ5 and BJ10 sensory attributes are equivalent to those of two commercial blend juices (FS and AFE) sold throughout Chile, indicating a high potential for consumer acceptance.

The stability of sensory attributes (color, appearance, consistency, aroma, and taste) of BJ5 and BJ10 relative to the BJ, as shown in Table 4, and the physical and chemical characteristics plotted in Table 3, agreed with the lack of significant differences (*p* > 0.05) among BJ, BJ5, and BJ10 in reducing sugars (Table 3) and taste (Table 4), which is a relevant attribute in fruit juices. In contrast, Lepaus et al. [32] established a significant decrease (*p* < 0.05) in the evaluation of aroma, flavor, overall acceptance, and purchase intent by the panelists after pasteurizing (90 °C for 30 s) an orange–carrot juice blend. A significant (α = 95%) effect was determined by Cendrowski et al. [33] on sensory preferences of different mixed juices prepared with rose fruits (*Rosa rugosa*) and apple or strawberry, treated at 85 °C/15 min and then packed into 100 mL glass bottles. In our research, the panelists found no significant difference (*p* > 0.05) in sensory attributes (Table 4) between the pasteurized blend juices (BJ5 and BJ10), and the two leading commercial blends (FS and AFE), both of which are also thermally treated products.

### 3.4. Shelf-Life of Pasteurized Blend Juice

The shelf-life stability of the pasteurized juices (BJ5 and BJ10) was evaluated over 8 weeks at 25 °C, with the results summarized in Table 5. In general, most physicochemical and microbiological parameters evaluated remained stable throughout the storage period, except for the total polyphenol content. Notably, both pasteurization treatments were effective in maintaining AMB and YM stability during all periods of storage.

During storage, a small but statistically significant decrease (*p* < 0.05) in moisture content was observed in both pasteurized blend juices (Table 5). However, at any given day, there was no significant difference in moisture between the two thermal treatments. According to Choo et al. [31] for noni juice, and Mandha et al. [44] for watermelon, pineapple, and mango juice, high moisture content (81.7–80.7%) of the blend juice allows microbial growth and chemical reactions during storage. Therefore, pasteurization at 80 °C for 1 to 15 min prevents microbial growth and ensures shelf stability.

The color difference (Δ*E)*, determined by comparing blend juice immediately after pasteurization and after each day of storage, showed values from 4.4 to 1.8, which are considered “very noticeable” and “medium” differences in color, according to the scale proposed by Adekunte et al. [26]. The *a** value (redness) decreased significantly (*p* < 0.05) for both treatments after two weeks of storage, which corresponds to the measure of red (+*a**) hue.

Soluble solid content increased significantly during storage (*p* < 0.05) for both pasteurized blend juices, which is consistent with the evolution of total soluble solids in watermelon juice pasteurized at 80 ± 2 °C for 10 and 15 min, and stored at 4 °C for 14 days [44]. These authors mentioned a similar behavior for roselle–mango juices attributed to the hydrolysis of polysaccharides into monosaccharides during storage.

During storage, the pH of the blend juice pasteurized for 5 min showed a significant increase (*p* < 0.05). In contrast, the pH of BJ10 remained stable, possibly due to the combined evolution of soluble solids and titratable acidity, which remained constant during storage.

The total polyphenol content was affected during storage, being significantly lower (*p* < 0.05) at the eighth week compared to blend juice immediately after pasteurization at 95 °C by 5 and 5 min treatments. Phenol degradation during storage is a well-documented phenomenon in fruit products and is consistent with the findings of Gonçalves et al. [15] for strawberry pulp and Mandha et al. [44] for pasteurized watermelon juice.

## 4. Conclusions

The blend juice formulated with a mixture of 70% BSG liquid fraction and 30% strawberry pulp was pasteurized at 95 °C for 5 and 10 min and immediately cooled. It exhibited excellent stability, showing no significant differences (*p* > 0.05) in physicochemical, microbiological, and sensory properties after the thermal treatments. Furthermore, the sensory characteristics, overall acceptability, and purchase intent of BJ5 and BJ10 were comparable to those of two commercial Chilean blend juices (FS and AFE), thus suggesting a high potential for consumer adoption and commercial success.

Throughout the 8-week storage period at 25 °C, both pasteurized blends (BJ5 and BJ10) maintained their color, chemical, and microbiological stability; however, significant changes (*p* < 0.05) were observed in moisture content, soluble solids, pH, and total polyphenol content. Nevertheless, both formulations remained free from aerobic mesophilic bacteria, yeasts, and molds throughout storage, confirming the microbiological stability achieved under both pasteurization conditions and their effectiveness for shelf-life preservation.

This study successfully demonstrates that the liquid fraction obtained from pressing brewer’s spent grain can be repurposed as a valuable dilution medium for strawberry pulp. The resulting novel blend juice, after pasteurization, exhibits an extended shelf-life of up to 8 weeks at 25 °C, maintaining its microbiological safety and key sensory attributes. This product not only represents a promising innovation for the food and gastronomic industries, but also contributes to the valorization of brewery by-products, aligning perfectly with the principles of a circular economy.

The consideration of BSG-LF variability is an essential part of the scientific method; variability assessment is the foundation of every experiment, especially when artisanal or industrial by-products are used. The present study was not designed to assess variability among different BSG batches; therefore, only a single batch was used. All material was obtained from the same mash to avoid batch-to-batch variation and ensure that the results were not influenced by extraneous factors. Nevertheless, we have planned a subsequent study to evaluate how different batches affect the characteristics of the pressed BSG liquid derived from both artisanal and industrial beer production. In that forthcoming work, we also intend to examine the various formulations used by an artisanal brewery and assess how differences in malt composition during mashing may influence the properties of the BSG liquid fraction obtained through pressing. This aspect is also of particular interest to us.

## Figures and Tables

**Figure 1 foods-14-04053-f001:**
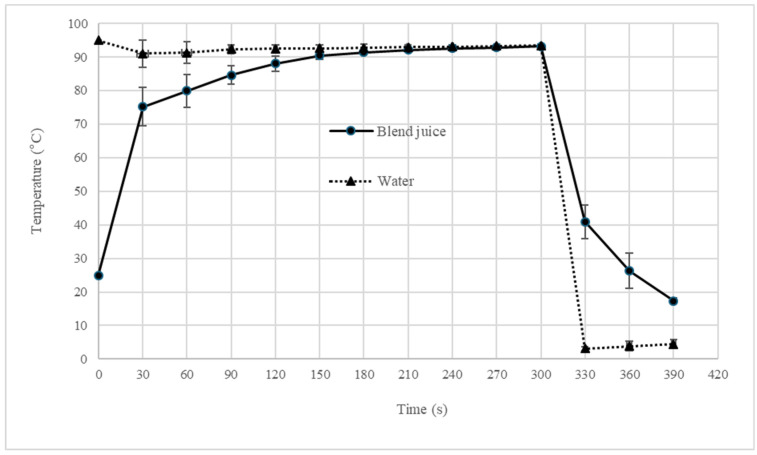
Temperature monitoring of the blend juice and bath water for pasteurization with a heating step of 5 min and subsequent cooling.

**Figure 2 foods-14-04053-f002:**
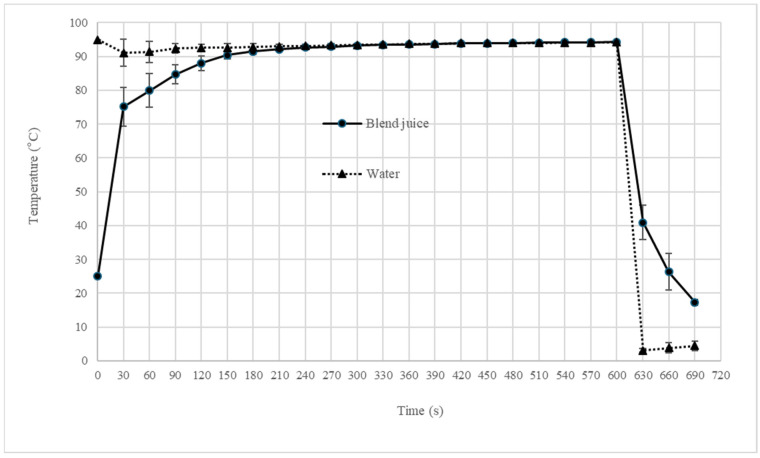
Temperature monitoring of the blend juice and bath water for pasteurization with a heating step of 10 min and subsequent cooling.

**Figure 3 foods-14-04053-f003:**
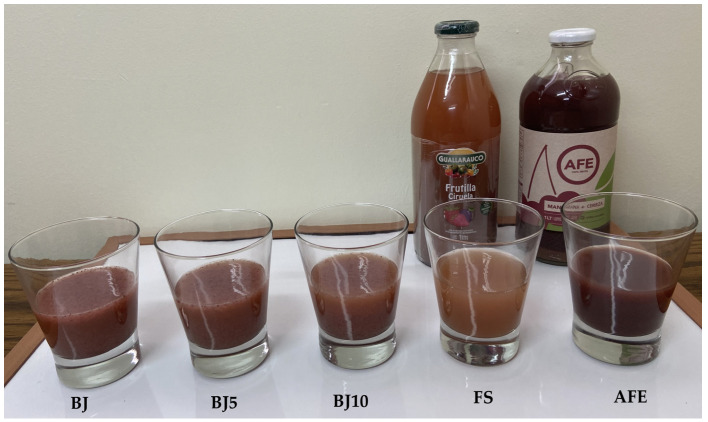
Blend juices used for sensory evaluation.

**Table 1 foods-14-04053-t001:** Moisture, color, chemical, and microbiological characteristics of BSG liquid fraction, strawberry pulp, and blend juice.

Parameter	BSG-LF	SP	BJ
Moisture content (%, w.b.)	87.0 ± 0.1 ^a^	69.2 ± 0.1 ^c^	82.4 ± 0.1 ^b^
Color			
*L**	43.7 ± 4.3 ^a^	30.1 ± 0.6 ^b^	34.4 ± 0.4 ^b^
*a**	0.2 ± 0.3 ^c^	15.9 ± 0.8 ^a^	8.9 ± 0.1 ^b^
*b**	12.9 ± 0.2 ^a^	11.4 ± 0.4 ^b^	10.6 ± 0.5 ^b^
Δ*E*	13.2 ± 2.9 ^a^	8.4 ± 0.4 ^a^	-
Soluble solid (°Brix)	13.5 ± 0.1 ^c^	29.1 ± 0.1 ^a^	17.2 ± 0.2 ^b^
Reducing sugar (g 100 g^−1^ d.m.)	65.6 ± 1.3 ^a^	20.9 ± 0.1 ^c^	47.4 ± 0.8 ^b^
pH	6.26 ± 0.09 ^a^	3.64 ± 0.16 ^c^	4.28 ± 0.03 ^c^
Titratable acidity (%)	0.065 ± 0.007 ^c^	0.663 ± 0.067 ^a^	0.243 ± 0.002 ^b^
TPC (mg GAE 100 g^−1^ d.m.)	169.4 ± 13.2 ^b^	256.6 ± 10.3 ^a^	181.2 ± 3.3 ^b^
Ascorbic acid (mg 100 g^−1^ d.m.)	<16.7	30.9 ± 3.2 ^a^	23.2 ± 1.5 ^b^
Microbiology			
AMB (log CFU mL^−1^)	4.8 ± 0.1 ^a^	Absent	3.5 ± 0.0 ^b^
YM (log CFU mL^−1^)	Absent	Absent	Absent

BSG-LF: BSG liquid fraction; SP: strawberry pulp; BJ: blend juice (70% BSG-LF—30% SP) before pasteurization; w.b.: wet basis; d.m.: dry matter; AMB: aerobic mesophilic bacteria; YM: yeast and mold; CFU: colony-forming unit. In each row, different letters indicate significant differences by Tukey’s test at *p* < 0.05 or by the t-Student test at *p* < 0.05 when two data were compared. Data are presented as mean ± standard deviation.

**Table 2 foods-14-04053-t002:** Sugar content (mg mL^−1^) in BSG liquid fraction, strawberry pulp, and blend juice before and after pasteurization.

Sample	Fructose	Glucose	Maltose	Raffinose
BSG-LF	n.d.	12.16 ± 0.16 ^d^	61.83 ± 0.72 ^a^	17.14 ± 0.39 ^a^
SP	28.50 ± 0.76 ^a^	31.05 ± 1.12 ^a^	n.d.	n.d.
BJ	11.22 ± 0.70 ^b^	23.87 ± 0.54 ^b^	34.58 ± 1.06 ^b^	11.84 ± 1.20 ^b^
BJ5	10.44 ± 0.42 ^b^	19.26 ± 0.46 ^c^	31.75 ± 0.96 ^c^	10.92 ± 0.86 ^b^
BJ10	11.19 ± 0.64 ^b^	20.82 ± 1.02 ^c^	34.50 ± 1.68 ^b^	11.48 ± 0.61 ^b^

BSG-LF: BSG liquid fraction; SP: strawberry pulp; BJ: blend juice (70% BSG-LF—30% SP) before pasteurization; BJ5 and BJ10: blend juice after pasteurization for 5 and 10 min at 95 °C, respectively; n.d.: not detected. In each column, different letters indicate significant differences by Tukey’s test at *p* < 0.05. Data are presented as mean ± standard deviation.

**Table 3 foods-14-04053-t003:** Moisture, color, chemical, and microbiological characteristics of the blend juice before and after pasteurization.

Parameter	BJ	BJ5	BJ10
Moisture content (%, w.b.)	82.4 ± 0.1 ^a^	81.9 ± 0.1 ^b^	81.7 ± 0.1 ^b^
Color			
*L**	34.4 ± 0.4 ^a^	33.5 ± 0.6 ^a^	33.9 ± 1.0 ^a^
*a**	8.9 ± 0.1 ^a^	8.7 ± 0.6 ^a^	8.7 ± 0.7 ^a^
*b**	10.6 ± 0.5 ^a^	9.4 ± 0.8 ^a^	9.9 ± 0.7 ^a^
Δ*E*	-	1.7 ± 0.5 ^a^	1.2 ± 0.3 ^a^
Soluble solid (° Brix)	17.2 ± 0.2 ^b^	17.7 ± 0.2 ^a^	18.0 ± 0.0 ^a^
Reducing sugar (g 100 g^−1^ d.m.)	47.4 ± 0.8 ^a^	48.9 ± 2.9 ^a^	49.4 ± 3.4 ^a^
pH	4.28 ± 0.03 ^a^	4.18 ± 0.02 ^ab^	4.14 ± 0.07 ^b^
Titratable acidity (%)	0.243 ± 0.002 ^a^	0.259 ± 0.031 ^a^	0.260 ± 0.030 ^a^
TPC (mg GAE 100 g^−1^ d.m.)	181.2 ± 3.3 ^ab^	172.8 ± 4.4 ^b^	186.8 ± 1.8 ^a^
Ascorbic acid (mg 100 g^−1^ d.m.)	23.2 ± 1.5 ^a^	27.9 ± 2.5 ^a^	26.7 ± 3.6 ^a^
Microbiology			
AMB (log CFU mL^−1^)	3.5 ± 0.0	Absent	Absent
YM (log CFU mL^−1^)	Absent	Absent	Absent

BJ: blend juice (70% BSG-LF—30% SP) before pasteurization; BJ5 and BJ10: blend juice after pasteurization for 5 and 10 min at 95 °C, respectively; w.b.: wet basis; d.m.: dry matter; AMB: aerobic mesophilic bacteria; YM: yeast and mold; CFU: colony-forming unit. In each row, different letters indicate significant differences by Tukey’s test at *p* < 0.05. Data are presented as mean ± standard deviation.

**Table 4 foods-14-04053-t004:** Sensory evaluation of the tested blend juice before and after pasteurization and two commercial fruit blend juices.

Sample	Color	Appearance	Consistency	Aroma	Taste	Overall Acceptance	Purchase Intent
BJ	7.2 ± 1.3 ^a^	6.5 ± 1.4 ^a^	6.9 ± 1.1 ^a^	6.8 ± 1.4 ^a^	7.2 ± 1.5 ^a^	6.7 ± 1.5 ^a^	6.7 ± 1.5 ^a^
BJ5	7.0 ± 1.5 ^a^	6.1 ± 1.8 ^a^	6.5 ± 1.4 ^a^	6.6 ± 1.6 ^a^	6.9 ± 1.6 ^ab^	6.7 ± 1.5 ^a^	6.5 ± 1.3 ^a^
BJ10	6.7 ± 1.6 ^ab^	6.0 ± 2.0 ^a^	6.7 ± 1.3 ^a^	6.5 ± 1.4 ^a^	6.5 ± 1.2 ^ab^	6.6 ± 1.4 ^a^	6.2 ± 1.7 ^a^
FS	5.7 ± 1.7 ^b^	5.8 ± 1.6 ^a^	6.8 ± 1.6 ^a^	6.5 ± 1.6 ^a^	5.8 ± 1.8 ^b^	6.2 ± 1.8 ^a^	5.6 ± 2.3 ^a^
AFE	6.4 ± 2.0 ^ab^	6.2 ± 2.2 ^a^	6.8 ± 1.5 ^a^	6.2 ± 1.4 ^a^	6.3 ± 1.6 ^ab^	6.2 ± 1.7 ^a^	5.7 ± 2.1 ^a^

BJ: blend juice (70% BSG-LF—30% SP) before pasteurization; BJ5 and BJ10: blend juice after pasteurization for 5 and 10 min at 95 °C, respectively; FS: strawberry–plum berry blend juice trademark Guallarauco (https://www.guallarauco.cl/, accessed on 14 October 2025); AFE: apple-cherry blend juice trademark AFE (https://www.jugoafe.cl, accessed on 14 October 2025). In each column, different letters indicate significant differences by Tukey’s test at *p* < 0.05. Data are presented as mean ± standard deviation.

**Table 5 foods-14-04053-t005:** Shelf-life at 25 °C of the blend juice after pasteurization at 90 °C by 5 and 10 min.

Parameter	Immediately After Pasteurized		Storage Time at 25 °C (Week)			
			1	2	4	8
Moisture content (%, w.b.)	BJ5	81.9 ± 0.1 ^aA^	81.6 ± 0.0 ^abA^	81.3 ± 0.2 ^bA^	81.7 ± 0.1 ^abA^	81.4 ± 0.2 ^bA^
	BJ10	81.7 ± 0.1 ^aA^	80.7 ± 0.2 ^cA^	80.9 ± 0.2 ^bcA^	81.4 ± 0.0 ^abA^	80.9 ± 0.0 ^bcA^
Color						
*L**	BJ5	33.5 ± 0.6 ^aA^	36.1 ± 2.6 ^aA^	34.3 ± 2.0 ^aA^	34.9 ± 1.7 ^aA^	34.6 ± 0.4 ^aA^
	BJ10	33.9 ± 1.0 ^bA^	37.3 ± 2.6 ^aA^	33.9 ± 0.7 ^bA^	33.6 ± 1.0 ^bA^	33.4 ± 0.6 ^bB^
*a**	BJ5	8.7 ± 0.6 ^aA^	8.5 ± 0.5 ^aA^	7.3 ± 0.8 ^bA^	6.5 ± 0.5 ^bB^	6.8 ± 0.5 ^bA^
	BJ10	8.7 ± 0.7 ^abA^	9.0 ± 0.7 ^aA^	7.6 ± 0.5 ^bcA^	7.1 ± 0.4 ^cA^	7.1 ± 0.6 ^cA^
*b**	BJ5	9.4 ± 0.8 ^aA^	10.5 ± 2.3 ^aA^	8.1 ± 0.5 ^aA^	9.7 ± 2.2 ^aA^	10.1 ± 1.6 ^aA^
	BJ10	9.9 ± 0.7 ^aA^	12.1 ± 2.6 ^aA^	9.3 ± 1.2 ^aA^	9.8 ± 1.7 ^aA^	9.8 ± 0.4 ^aA^
Δ*E*	BJ5	-	2.9 ± 2.5 ^aA^	2.7 ± 1.4 ^aA^	3.6 ± 2.1 ^aA^	3.2 ± 0.7 ^aA^
	BJ10	-	4.4 ± 2.6 ^aA^	1.8 ± 0.9 ^aA^	2.5 ± 0.7 ^aA^	2.0 ± 1.0 ^aA^
Soluble solid (°Brix)	BJ5	17.7 ± 0.2 ^bcA^	17.5 ± 0.0 ^cB^	18.0 ± 0.1 ^abB^	18.2 ± 0.3 ^aA^	18.3 ± 0.2 ^aB^
	BJ10	18.0 ± 0.0 ^dA^	19.1 ± 0.1 ^aA^	18.9 ± 0.1 ^bA^	18.5 ± 0.2 ^cA^	19.0 ± 0.1 ^abA^
Reducing sugar (g 100 g^−1^ d.m.)	BJ5	48.9 ± 2.9 ^aA^	46.2 ± 0.4 ^aA^	48.0 ± 5.2 ^aA^	49.1 ± 0.9 ^aA^	47.2 ± 0.7 ^aB^
	BJ10	49.4 ± 3.4 ^aA^	49.3 ± 3.4 ^aA^	42.4 ± 0.2 ^bA^	48.6 ± 1.7 ^aA^	49.4 ± 0.3 ^aA^
pH	BJ5	4.18 ± 0.02 ^bA^	4.07 ± 0.01 ^cB^	4.19 ± 0.01 ^bA^	4.17 ± 0.01 ^bB^	4.27 ± 0.01 ^aA^
	BJ10	4.14 ± 0.07 ^aA^	4.17 ± 0.01 ^aA^	4.17 ± 0.01 ^aA^	4.24 ± 0.01 ^aA^	4.27 ± 0.01 ^aA^
Titratable acidity (%)	BJ5	0.259 ± 0.031 ^aA^	0.254 ± 0.001 ^aA^	0.247 ± 0.001 ^aA^	0.242 ± 0.002 ^aA^	0.252 ± 0.001 ^aA^
	BJ10	0.260 ± 0.030 ^aA^	0.280 ± 0.001 ^aB^	0.246 ± 0.002 ^aA^	0.250 ± 0.000 ^aA^	0.251 ± 0.001 ^aA^
TPC (mg GAE 100 g^−1^ d.m.)	BJ5	172.8 ± 4.4 ^aB^	170.0 ± 2.2 ^aB^	147.8 ± 0.3 ^bB^	149.0 ± 7.8 ^bB^	134.0 ± 3.7 ^cA^
	BJ10	186.8 ± 1.8 ^abA^	195.7 ± 5.1 ^aA^	160.1 ± 0.0 ^cA^	175.9 ± 7.5 ^bA^	136.2 ± 7.8 ^dA^
Ascorbic acid (mg 100 g^−1^ d.m.)	BJ5	27.9 ± 2.5 ^aA^	23.5 ± 1.3 ^abA^	21.1 ± 0.9 ^bA^	22.9 ± 0.9 ^abA^	22.0 ± 1.5 ^abA^
	BJ10	26.7 ± 3.6 ^aA^	22.2 ± 1.2 ^aA^	23.1 ± 1.5 ^aA^	23.2 ± 0.6 ^aA^	21.4 ± 0.3 ^aA^
Microbiology						
AMB (log CFU mL^−1^)	BJ5	Absent	Absent	Absent	Absent	Absent
	BJ10	Absent	Absent	Absent	Absent	Absent
YM (log CFU mL^−1^)	BJ5	Absent	Absent	Absent	Absent	Absent
	BJ10	Absent	Absent	Absent	Absent	Absent

BJ5 and BJ10: blend juice after pasteurization for 5 and 10 min at 95 °C, respectively; w.b.: wet basis; d.m.: dry matter; AMB: aerobic mesophilic bacteria; YM: yeast and mold; CFU: colony-forming unit. For each parameter, different uppercase letters indicate significant differences by the *t*-Student test at *p* < 0.05 between BJ5 and BJ10 at the same storage time (columns), while different lowercase letters indicate significant differences based on Tukey’s test at *p* < 0.05 (rows). Data are presented as mean ± standard deviation.

## Data Availability

The original contributions presented in the study are included in the article; further inquiries can be directed to the corresponding authors.

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
