# Peer review of "Valorization of Brewer’s Spent Grain Liquid Fraction for the Development of a Pasteurized Strawberry-Based Blend Juice"

_foods, 2025, doi:10.3390/foods14234053_

Round 1
Reviewer 1 Report
Comments and Suggestions for Authors
This research is a highly practical study with significant research value. However, some parts still need to be improved.
1. What components of liquid fraction of Brewer’s spent grain contain?
2. Are there any differences between the two types of pasteurization time treatments?
3. The quality of this research differs somewhat from that of the products sold in the market. Especially in terms of appearance, how should it be improved?
4. The changes in quality and physical and chemical components during storage also need to be demonstrated.
5. The article needs to include some photos of real products to enable readers to have a more intuitive understanding.
Comments on the Quality of English LanguageIt should be improved.
Author Response
We sincerely appreciate your time and effort in reviewing our manuscript and for the valuable comments provided. We have carefully considered each point, addressed them in detail, and incorporated the corresponding revisions where appropriate. Your thoughtful feedback has significantly contributed to improving the quality and clarity of the manuscript. We hope our responses and clarifications satisfactorily address your concerns. Thank you once again for your constructive input.
The line numbers now mentioned in the responses, which indicate the modifications made to the manuscript, correspond to the revised version.
- What components of liquid fraction of Brewer’s spent grain contain?
According to the proximate composition of the Brewer’s spent grain liquid fraction (BSG-LF) used in this study, it contained fat (<0.15%), protein (0.4%), ash (<0.043%), and nitrogen-free extract (12.5%), with sodium levels below 4 mg/100 g. (Lines 263 - 265). These compositional data were considered nonessential for inclusion in the manuscript, but now was included. Regarding carbohydrates, the BSG-LF comprised glucose, maltose, and raffinose at concentrations of 12.16 ± 0.16, 61.83 ± 0.72, and 17.14 ± 0.39 mg/mL, respectively.
- Are there any differences between the two types of pasteurization time treatments?
The two pasteurization treatments applied to the blended juice (BJ) differed only in duration—5 and 10 minutes—while all other conditions remained identical, as described in Section 2.2, Blended Juice Pasteurization (Lines 122–130). Their effects are illustrated in Figures 1 and 2, and the corresponding results, including statistical analyses, are summarized in Tables 2 to 4. Based on these data, significant differences between the 5- and 10-minute pasteurization treatments were observed only for maltose content (Table 2) and total polyphenol content (Table 4). Therefore, under the conditions employed in this study, extending pasteurization from 5 to 10 minutes had negligible overall impact on the properties of the blended juice.
- The quality of this research differs somewhat from that of the products sold in the market. Especially in terms of appearance, how should it be improved?
As shown in Table 3, no statistically significant differences in appearance were found between BJ, BJ5, and BJ10 compared with commercial products available on the market—specifically, the strawberry–cherry blend juice (Fundo Sofruco, https://www.larosasofruco.cl) and the apple–cherry blend juice (AFE, https://www.jugoafe.cl). Nevertheless, the visual quality of the BJ formulated in this study could be further enhanced to increase its market appeal. The primary avenue for improvement lies in better preservation of color intensity and blend homogeneity during processing and storage, aspects that will be addressed in future research.
- The changes in quality and physical and chemical components during storage also need to be demonstrated.
Table 5 presents the evolution of the physical and chemical parameters of the blended juice during storage, along with the corresponding statistical analyses. These results, which include variations in moisture content, soluble solids, pH, and total polyphenol content, are thoroughly discussed in lines 485–509, that clarifies the impact of storage. As summarized in the conclusion: “During storage at 25 °C for 8 weeks, BJ5 and BJ10 maintained their color, chemical, and microbiological stability; however, significant changes (p < 0.05) were observed in moisture content, soluble solids, pH, and total polyphenol content. Both formulations remained free from aerobic mesophilic bacteria, yeasts, and molds throughout storage, confirming the microbiological stability achieved under both pasteurization conditions”. We believe that the data presented in Table 5, along with the related discussion, clearly demonstrate which parameters were affected and which remained stable, thereby addressing the changes in quality over the storage period. Then, data presented in this table effectively illustrate which quality parameters were significantly affected (or unaffected) after pasteurization and storage.
- The article needs to include some photos of real products to enable readers to have a more intuitive understanding.
Lines 245–247 provide links to the websites of the commercial products (FS: strawberry–cherry blend juice, Fundo Sofruco, https://www.larosasofruco.cl; AFE: apple–cherry blend juice, AFE, https://www.jugoafe.cl), allowing readers to view the products directly. While including actual photographs in the manuscript would indeed enhance readers’ intuitive understanding, the feasibility of adding images depends on the journal’s layout and space constraints. We will verify with Foods whether there is provision for incorporating such figures.
Comments on the Quality of English Language
It should be improved.
The English language of the manuscript has been reviewed; if a formal language-editing service is required, we will arrange it.

Reviewer 2 Report
Comments and Suggestions for Authors
Dear authors,
the content of the article “Valorization of Brewer’s Spent Grain Liquid Fraction for the Development of a Pasteurized Strawberry-Based Blend Juice” is original because it explores the possibility of using waste that has a significant environmental impact, but which also has excellent nutraceutical potential, to produce a beneficial consumer product.
The experiment appears to have been well organised and conducted, involving a variety of chemical and physical tests that can also be used to predict the product's shelf life and consumer acceptability.
However, as mentioned in the comments below, the microbiological analyses carried out do not seem exhaustive enough to guarantee consumer safety. Below are some suggestions, some of which are intended to help the authors improve their presentation and others of which are more substantial.
Overall, the article can be considered suitable for publication after minor revisions.
Thank you
Best regards

Author Response
We sincerely appreciate your time and effort in reviewing our manuscript and for the valuable comments provided. We have carefully considered each point, addressed them in detail, and incorporated the corresponding revisions where appropriate. Your thoughtful feedback has significantly contributed to improving the quality and clarity of the manuscript. We hope our responses and clarifications satisfactorily address your concerns. Thank you once again for your constructive input.
The line numbers now mentioned in the responses, which indicate the modifications made to the manuscript, correspond to the revised version.
2.4.9. Microbiology Determination (line 204-210)
- Do the two categories of microorganisms analysed provide sufficient guarantees for the safety of the product? Similar articles, such as the one cited in relation to heat treatments (Xue et al. [22]) and EL Moutaouakil et al. [41], have analysed coliforms in addition to aerobic mesophilic bacteria (AMB) and yeast and mould (YM). For instance, the relevant EU legislation states that coliforms and Enterobacteriaceae must be determined in certain foods, including fruit juices. The authors explain their reasoning behind this choice.
The determination of coliforms, total coliforms, and Enterobacteriaceae (including Escherichia coli) is indeed required by food safety regulations in most countries, including Chile, to ensure the safety of products marketed nationally and internationally. In the studies by Xue et al. [22] and EL Moutaouakil et al. [41], coliform analysis was included because the research focused on commercial products produced in processing facilities, where contamination from multiple sources is possible. In contrast, the present study aims to evaluate shelf life under controlled laboratory conditions, where samples are handled only during analysis and there is no risk of external contamination. Therefore, monitoring spoilage microorganisms—specifically aerobic mesophilic bacteria, yeasts, and molds—is sufficient for assessing product stability and quality in this context.
- Lines 282-285: In my opinion, drawing comparisons between these and matrices that differ greatly from the liquid fraction of brewers' grains could cause confusion among readers. This sentence could be deleted.
We recognize that the sentence in Lines 282–285 could potentially confuse readers due to the differences between the compared matrices and the liquid fraction of brewers’ spent grains. However, we believe retaining it is useful to illustrate that, although polyphenols are present in BSG-LF, their total content is generally lower than in other matrices (Lines 305-306). This comparison provides context for interpreting the polyphenol levels observed in our study.
Similarly, we acknowledge that the sentence in Lines 300–304 could potentially confuse readers due to the differences between the compared matrices and the liquid fraction of brewers’ spent grains. However, we believe retaining it is useful to illustrate that, although polyphenols are present in BSG-LF, their total content is generally lower than in other matrices (Lines 305-306). This comparison provides context for interpreting the polyphenol levels observed in our study.
- Line 321: What does LB-BSG stand for? If I'm not mistaken, it has never been mentioned before.
You are correct; “LB-BSG” is a typographical error. The correct term is BSG-LF, and this has been corrected throughout the text.
- Lines 382-412: In order to give more prominence to this analysis, which is a strength of the article, the description of the sensory analysis results should be included in a different sub-section (3.3 or as the authors prefer). This will also prevent interruption to the description of the results of chemical-physical determinations.
One way to organise the paragraphs in this section would be as follows:
3.1. Characterization of BSG Liquid Fraction, Strawberry Pulp and Blend Juice
3.2. Effect of Pasteurization on Blend Juice
3.3. 3.3. Sensory Evaluation of Pasteurized Blend Juice
3.4. Shelf Life of Pasteurized Blend Juice
We agree with the suggestion and have created a separate subsection (3.3) dedicated to the sensory evaluation of the pasteurized blend juice. This reorganization highlights the sensory analysis as a key strength of the study and prevents interruption of the description of the physicochemical results. The revised structure of Section 3 is now as follows: 3.1. Characterization of BSG Liquid Fraction, Strawberry Pulp, and Blend Juice; 3.2. Effect of Pasteurization on Blend Juice; 3.3. Sensory Evaluation of Pasteurized Blend Juice; 3.4. Shelf Life of Pasteurized Blend Juice. (Lines 454-489)
- Lines 453-454: “For one parameter, in each column, different capital letters indicate significant differences determined by the t-Student test at p < 0.05”. I'm not sure I understand. Do capital letters refer to a comparison made along the column (i.e. between different parameters at the same storage time) or within the same row?
The comparison is made for the same parameter at each storage time, between the two samples, BJ5 and BJ10. For example, for moisture content immediately after pasteurization, BJ5 is compared with BJ10, which do not differ significantly in Table 5. In response to your comment, the sentence has been revised for clarity as follows:
“For each parameter, different uppercase letters indicate significant differences by the t-Student (p<0.05) between BJ5 and BJ10 at the same storage time (columns)” (Lines 499-501)

Reviewer 3 Report
Comments and Suggestions for Authors
Given the circular economy trend, the topic of this paper is novel. Still, there are some practical lacks of using BSG.
Commonly, the yeast and molds grow quickly in a favorable medium. And BSG is a proper medium for microorganisms to grow. Specify the moment of collecting the samples of BSG from the brewery, how long was it kept and in which conditions until the moment of analysis. It is almost impossible to have absent microorganisms in BSG.
What was the composition of BSG? Decide about its moisture content.
From which malt assortments (including producer details) was BSG derived?
Author Response
We sincerely appreciate your time and effort in reviewing our manuscript and for the valuable comments provided. We have carefully considered each point, addressed them in detail, and incorporated the corresponding revisions where appropriate. Your thoughtful feedback has significantly contributed to improving the quality and clarity of the manuscript. We hope our responses and clarifications satisfactorily address your concerns. Thank you once again for your constructive input.
The line numbers now mentioned in the responses, which indicate the modifications made to the manuscript, correspond to the revised version.
- Given the circular economy trend, the topic of this paper is novel. Still, there are some practical lacks of using BSG. Commonly, the yeast and molds grow quickly in a favorable medium. And BSG is a proper medium for microorganisms to grow. Specify the moment of collecting the samples of BSG from the brewery, how long was it kept and in which conditions until the moment of analysis. It is almost impossible to have absent microorganisms in BSG.
The BSG was collected directly by the research group from the Birrell craft brewery immediately after mashing, when the temperature reaches approximately 70 °C, likely inactivating molds and yeasts. The batch used throughout the study was stored in washed and sanitized plastic boxes, which were only opened when filling them with BSG. Transport to the laboratory took approximately one hour, and the procedure described in Section 2.1, BSG Liquid Fraction, Strawberry Pulp, and Blend Juice (Lines 107–112), was followed. As shown in Table 1, aerobic mesophilic bacteria were present, whereas yeasts and molds were absent. This observation is consistent with the reviewer’s comment, indicating that while wet BSG inherently contains microorganisms, their transfer to the liquid fraction occurs during the pressing step, as described at item 2.1. Furthermore, pasteurization of the blend juices reduced microbial load, thereby preventing the growth of spoilage-causing microorganisms.
- What was the composition of BSG? Decide about its moisture content.
The proximate composition of the BSG used in this study, on a wet basis, was as follows: moisture 72.34%, fat 1.65%, protein 5.13%, crude fiber 3.19%, ash 0.68%, nitrogen-free extract 17.01%, and sodium 2.3 mg/100 g (Lines 104-106). Additional details regarding the wet BSG can be found in article number 8 (Line 586).
The BSG was derived from malt supplied by Maltexco, a company based in Temuco, Chile (https://www.maltexco.com/web/), which provides malt to most craft breweries in the Temuco region, including Birrell Ltda., a craft brewery in Villarrica, Araucanía Region, Chile. The specific malt variety used to produce the BSG was a blend of Patagonia pilsner (50%) and caramel malt (50%) provided by Maltexco.

Reviewer 4 Report
Comments and Suggestions for Authors
A relevant study was conducted evaluating the practical valorization of the Brewer’s Spent Grain liquid fraction. However, it remains unclear why only one ratio of this by-product to strawberry pulp was selected — a stronger scientific rationale for this choice should be provided. Most of the analyses focus on basic technological properties of the product; therefore, it is recommended to place greater emphasis on the scientific contribution and innovative aspects of the work.
Line 112: Please provide a more detailed description of how the BSG pressing was performed, including specific conditions (e.g., equipment used, applied pressure, duration, and temperature).
Figures 1 and 2 do not appear to provide additional scientific value or new information. As no novel method is being developed or described, these figures seem unnecessary and could be removed without affecting the quality or clarity of the manuscript.
Table 2: The obtained results should be discussed in more depth, particularly regarding the impact of pasteurization on the saccharide profile. Please explain why the concentration of some compounds increased after longer pasteurization, while the overall saccharide content decreased. Additionally, clarify how the BJ sample, which contained only 30% SP, exhibited such high levels of fructose and glucose but notably low maltose content.
Lines 383–384: This information should also be included in the Methods section, specifying the manufacturers and main ingredients to ensure clarity and comparability of the obtained results.
Lines 393–394: This information should be moved to the Methods section and expanded to include the age range of the panelists, as well as the number of male and female participants involved in the evaluation.
Line 401: Please verify the reference — it appears that this should refer to Table 3 instead Lines 400–401: Please clarify the meaning of this sentence. As currently written, it is unclear how the maintenance of sensory attributes in BJ5 and BJ10 samples is said to “agree with the sugar composition.” Explain the relationship you intended to highlight between sensory characteristics and sugar profile. Lines 436–440: Please explain the reason behind the observed significant decrease in pH in your samples.Author Response
We sincerely appreciate your time and effort in reviewing our manuscript and for the valuable comments provided. We have carefully considered each point, addressed them in detail, and incorporated the corresponding revisions where appropriate. Your thoughtful feedback has significantly contributed to improving the quality and clarity of the manuscript. We hope our responses and clarifications satisfactorily address your concerns. Thank you once again for your constructive input.
The line numbers now mentioned in the responses, which indicate the modifications made to the manuscript, correspond to the revised version.
- A relevant study was conducted evaluating the practical valorization of the Brewer’s Spent Grain liquid fraction. However, it remains unclear why only one ratio of this by-product to strawberry pulp was selected — a stronger scientific rationale for this choice should be provided. Most of the analyses focus on basic technological properties of the product; therefore, it is recommended to place greater emphasis on the scientific contribution and innovative aspects of the work.
From a practical standpoint, the manufacturer of the frozen strawberry pulp Varfel (https://www.varfel.cl/) indicates on the product label that the preparation method is to use 333 g of pulp and dilute with water to 1 L. As mentioned in the manuscript, the blend juice (BJ) was established during preliminary experiments with a proportion of 70% BSG liquid fraction and 30% strawberry pulp. Different proportions affected physical, chemical, and organoleptic characteristics, including color, soluble solids, reducing sugars, titratable acidity, and total phenolic content (TPC). Finally, since the BSG-LF also contributes solids, which pure water does not, a 30% proportion of strawberry pulp was considered appropriate to achieve the desired composition.
- Line 112: Please provide a more detailed description of how the BSG pressing was performed, including specific conditions (e.g., equipment used, applied pressure, duration, and temperature).
The BSG was pressed using a manual stainless steel press with a 2 L container and mechanical screw, similar to those traditionally employed for fruit, such as stainless steel hydraulic apple wine presses. Pressing was conducted at room temperature (23–25 °C) for 5 minutes (Line 117). As the press is manually operated, no pressure gauge was available.
- Figures 1 and 2 do not appear to provide additional scientific value or new information. As no novel method is being developed or described, these figures seem unnecessary and could be removed without affecting the quality or clarity of the manuscript.
We agree with the reviewer that the thermal treatment method is widely used in the food industry and is not novel. However, we consider that Figures 1 and 2 effectively demonstrate the actual thermal behavior of the blended juice, which differs from the nominal temperature–time combinations of 95 °C for 5 and 10 minutes described in Section 2.2. Therefore, these figures help to illustrate how the temperature–time regimen was actually applied.
- Table 2: The obtained results should be discussed in more depth, particularly regarding the impact of pasteurization on the saccharide profile. Please explain why the concentration of some compounds increased after longer pasteurization, while the overall saccharide content decreased. Additionally, clarify how the BJ sample, which contained only 30% SP, exhibited such high levels of fructose and glucose but notably low maltose content.
Initially, when performing a mass balance for the 70% BSG-LF and 30% SP proportion, the results are consistent for all sugars listed in Table 2. In the BJ blend, the fructose originates solely from the strawberry pulp, which explains why it decreases to approximately one-third of the initial value, considering the mass ratio. Similar decreases are observed for glucose, maltose, and raffinose. However, the concentrations of these sugars remain nearly constant after pasteurization for 5 minutes (BJ5) and 10 minutes (BJ10), compared to before pasteurization (BJ). (Lines 403-409)
Regarding fructose and glucose, the measured values are slightly higher than the theoretical mass balance calculations, but not excessively so, considering the 30% proportion of SP. This may be due to more effective extraction of sugars from the fruit when mixed with BSG-LF. The same reasoning applies to maltose, which is lower in BJ relative to BSG-LF due to the 70% proportion. Additionally, when mixed with the pulp (SP), probably changes in pH and acidity (Table 1) may promote partial degradation of maltose in glucose. (Lines 411-417)
- Lines 383–384: This information should also be included in the Methods section, specifying the manufacturers and main ingredients to ensure clarity and comparability of the obtained results.
In Section 2.6 of the Methods, information on the manufacturers and main ingredients of both commercial juices was included. The ingredients of the 1-liter Strawberry-Cherry Blend Juice (trademark Fundo Sofruco), as stated on the label, are concentrated grape juice, strawberry, and plum pulp. In contrast, the 1-liter Apple-Cherry Blend Juice (trademark AFE), also listed on the label, is made exclusively from green apples and selected fresh cherries, without any additional additives (Lines 245-251).
- Lines 393–394: This information should be moved to the Methods section and expanded to include the age range of the panelists, as well as the number of male and female participants involved in the evaluation.
Section 2.6 of the Methods was updated to include the requested information. The panelists ranged in age from 20 to 62 years (mean age 37.9 ± 13.4 years), including 13 females and an equal number of males (Lines 236-238).
- Line 401: Please verify the reference — it appears that this should refer to Table 3 instead
Thank you for pointing that out. In the updated version of the manuscript, the correct reference is Table 4.
- Lines 400–401: Please clarify the meaning of this sentence. As currently written, it is unclear how the maintenance of sensory attributes in BJ5 and BJ10 samples is said to “agree with the sugar composition.” Explain the relationship you intended to highlight between sensory characteristics and sugar profile.
It is explained and replaced with the following text: “agreed with the no significant difference (p > 0.05) among BJ, BJ5, and BJ10 in reducing sugars (Table 3) and taste (Table 4), which is a relevant attribute in fruit juices.” (Lines 477-479
- Lines 436–440: Please explain the reason behind the observed significant decrease in pH in your samples.
The standard deviation of pH (Table 3 in the revised manuscript) for BJ10 is higher than that for BJ and BJ5, which contributes to the statistical analysis indicating a significant difference (p < 0.05). This results in a decrease in pH for BJ10 compared to BJ, but not compared to BJ5. From a chemical standpoint, this is consistent with a slight (non-significant) increase in titratable acidity in BJ10 compared to BJ, which could be attributed to the release of organic acids from the strawberry, promoted by the effect of heat on the cell membranes. (Lines 444-446)

Round 2
Reviewer 1 Report
Comments and Suggestions for Authors
This manuscript needs some pictures for the new products, compared with the commercial product.
Comments on the Quality of English LanguageIt should be improved.
Author Response
Thank you again for requesting a picture.
Accordingly, Figure 3 (Line 477), which presents the new products in comparison with the commercial product, has been added.
The English language will be further refined once the remaining reviewers have submitted their feedback.
